# StagFormer: Time Staggering Decoder-only Transformers

**Dylan Cutler**
Google
dylancutler@google.com

**Arun Kandoor**
Google Research
akandoor@google.com

**Nishanth Dikkala**
Google Research
nishanthd@google.com

**Nikunj Saunshi**
Google Research
nsaunshi@google.com

**Xin Wang**
Google Research
wanxin@google.com

**Rina Panigrahy**
Google Research
rinap@google.com

## Abstract

Decoding in a Transformer based language model is inherently sequential as a token's embedding needs to pass through all the layers in the network before the generation of the next token can begin. In this work, we propose a new architecture StagFormer (Staggered Transformer), which staggers execution along the sequence axis and thereby enables partial parallelization of the decoding process along the depth of the model. We achieve this by breaking the dependency of the token representation at time step $i$ in layer $l$ upon the representations of tokens until time step $i$ from layer $l-1$. Instead, we stagger the execution and only allow a dependency on token representations until time step $i-1$. The later sections of the Transformer still get access to the "rich" representations from the prior section but only from those token positions which are one time step behind. StagFormer allows for different sections of the model to be executed in parallel yielding a speedup in decoding while being quality neutral in our simulations. We also explore many natural extensions of this idea. We present how weight-sharing across the different sections being staggered can be more practical in settings with limited memory. We explore the efficacy of using a bounded window attention to pass information from one section to another which helps drive further latency gains for some applications. We also explore the scalability of the staggering idea over more than 2 sections of the Transformer enabling more parallelism. Finally, we show how one can approximate a recurrent model during inference using weight-sharing. This variant can lead to substantial gains in quality for short generations while being neutral in its latency impact.

## 1 Introduction

The Transformer architecture (Vaswani et al., 2017) has seen tremendous success as the primary backbone for language models (Chowdhery et al., 2022; Hoffmann et al., 2022; Brown et al., 2020). It lends itself particularly well for causal language modeling by allowing efficient, highly parallelized training over large datasets. Moreover, the model can be efficiently partitioned across multiple devices (Pope et al., 2022) enabling model parallelism across machines. However, it is well known that, during inference, decoding from a Transformer is an inherently sequential task. This task becomes more expensive when trying to decode long sequences due to the cost of attention, which requires computation that scales linearly with respect to sequence length for each additional token.

Consequently, there has been a significant body of research which tries to make inference from Transformers more efficient in practice. Speculative decoding, local attention and other efficient attention variants (Tay et al., 2022), KV cache optimizations, blockwise parallel decoding (Stern et al., 2018) etc. are a few such works. However, there haven't been many works which try to tackle the sequentiality imposed by the depth of the Transformer. Depth,

while known to be essential for the strong performance of Transformers (Raffel et al., 2023; Zhao et al., 2023; Ye et al., 2024), introduces a proportional cost in terms of decoding latency.

In this work, we take a look at how we can introduce some degree of parallel execution along the depth axis of a causally trained Transformer language model while decoding.

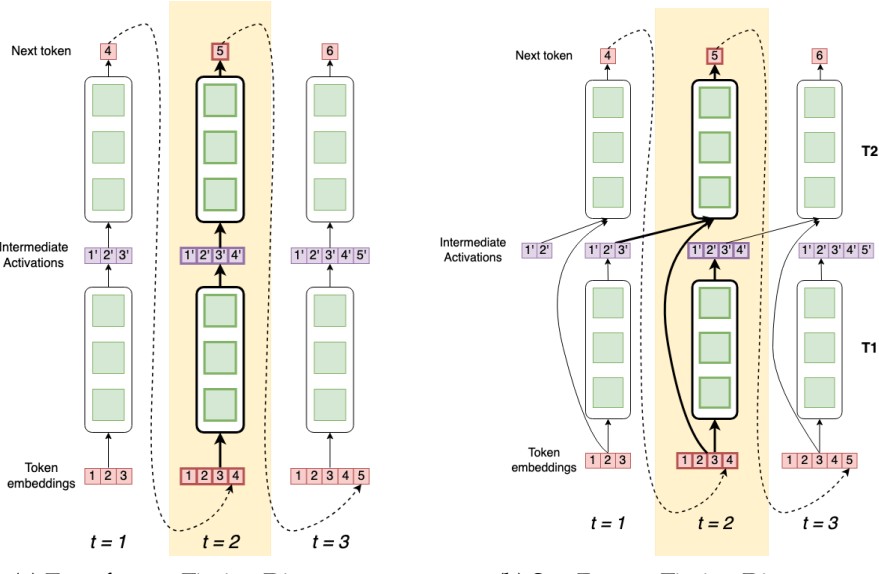

(a) Transformer Timing Diagram        (b) StagFormer Timing Diagram

Figure 1: Depiction of forward pass in a standard Transformer compared with that of StagFormer. Note that in StagFormer, the data dependency in a given time step has been broken for the two stacks T1 and T2.

We introduce StagFormer (*Staggered Transformer*), a novel Transformer variant which breaks the sequential dependency of the upper layers on the lower layers by *staggering* the time dependency of token embeddings passed from the lower layers to the upper layers. In particular, we devise a mechanism by which, at time step $i$, the upper layers of the model use the rich representations of tokens computed by earlier layers only until time step $i - 1$. Note that in a standard Transformer this dependency is allowed until time step $i$. We show how one can train and decode efficiently while matching the standard Transformer's quality using our architecture.

We demonstrate the performance of the StagFormer architecture on language modeling on the Pile dataset (Gao et al., 2020). We show that we can get significant latency savings during decode due to parallel execution of different parts of the Transformer stack while being neutral in quality. Finally, we also explore some natural variants and generalizations of the StagFormer architecture and demonstrate their efficacy as well for language modeling. We include a thorough downstream task evaluation for our trained language models across a wide suite of tasks involving summarization, reasoning, and factuality among others.

## 1.1 Related Work

The Transformer was originally proposed in the seminal work of Vaswani et al. (2017). Decoder-only language modeling using the Transformer was originally proposed by Radford (2018) and has since become a standard backbone to many frontier language models today.

There has been an enormous body of research dedicated towards making Transformer training or inference more efficient (Tay et al., 2022). These involve approaches which focus on pre-training such as distillation (Xu et al., 2024), layer stacking (Panigrahi et al., 2024), Alternating-updates (Baykal et al., 2024), Matryoshka Transformer (Kusupati et al., 2022) among others. Quantization (Xiao et al., 2023) has been another widely successful approach

Table 1: StagFormer vs Standard Transformer: Pretrained on the Pile dataset for 300B tokens. We see that StagFormer is able to match or outperform the 36 layer baseline model. Even though the time staggering should give some quality loss, this is being offset by the additional cross-attention weights.

| Model | Pile Pplx. | HellaSwag | ARC-E | ARC-C | WinoGrande | SuperGLUE | SQuADv2 | GEM-XSum rouge2 | Avg. |
|---|---|---|---|---|---|---|---|---|---|
| Baseline (18L) 1.6B params | 4.026 | 49.8 | 60.1 | 31.8 | 53.4 | 59.3 | 31.8 | 0.9 | 41.0 |
| Baseline (36L) 2.8B params | 3.780 | 53.3 | 66.7 | 34.6 | 60.4 | 62.1 | 36.3 | 1.6 | 45.0 |
| StagFormer $p = 2$ (2 x 18L Stacks) 2.9B params | **3.756** | **58** | **66.8** | **36.3** | **60.5** | 61.3 | **44.4** | 1.5 | **47.0** |

at speeding up inference of language models. There have been other approaches specifically focused on improving the decoding speed from language models such as speculative decoding and related works (Leviathan et al., 2023; Sun et al., 2024; Santilli et al., 2023).

There has also been a huge body of work focusing on making the self-attention more efficient. Some of these works have introduced the idea of introducing a form of recurrence mechanism into models, such as Transformer-XL and State Space Models (SSMs) like Mamba (Dai et al., 2019; Gu et al., 2022; Gu & Dao, 2024). Block-Recurrent Transformers use cross-attention to introduce a per-layer recurrence mechanism into Transformer networks (Hutchins et al., 2022).

More closely related to our effort are works such as Medusa (Cai et al., 2024) which uses parallel heads to decode multiple tokens ahead at once, Staircase Attention (Ju et al., 2022) which uses a similar idea of staggering attention window context as we advance deeper into the Transformer stack. However, they mainly explore a variant of the idea which allows one to bring in the benefits of RNNs rather than efficiency gains, our main focus here.

Our shared-weight variant of StagFormer, introduced in Section 4.1, is closely related to the idea of a *looped* Transformer, where the hidden activation signals are sent through the layers of the network multiple times (Dehghani et al., 2018; Giannou et al., 2023; Gatmiry et al., 2024). Part of the intuition behind looping is that the lower layers of a network can reuse the more-information-rich activations from layers later in the same network in the next iteration of the loop to create higher quality representations. A key difference in our method from looping is that it breaks the strict data-dependency on each prior loop, allowing for parallel execution of different passes through the network.

## 2 Staggered Transformers (StagFormer)

In this section we describe our Staggered Transformer (StagFormer) architecture. We begin with a brief background on a decoder-only language models based on the standard Transformer, the backbone for most state-of-the-art language models today.

**Language Modeling with the Transformer** A Transformer of depth $\ell$ is a sequence-to-sequence model which takes in a token sequence of length $N$ and generates an output sequence of length $N$. The tokens are each first mapped to a $d$-dimensional representation using an embedding layer. Positional information may also be combined into the embedding at this stage. Denote the token embeddings so obtained by $\mathbf{t}_0^{1,\dots,N}$. Then, these representations are progressively modified by applying a sequence of Transformer layers, $L_1, \dots, L_\ell \colon \mathbb{R}^d \to \mathbb{R}^d$ iteratively: $\mathbf{t}_j^{1,\dots,N} = L_j\left(\mathbf{t}_{j-1}^{1,\dots,N}\right)$ for $j \in \{1, \dots, \ell\}$. Each layer $L_j$ consists of two main operations: (a) self-attention which combines information across the different token embeddings and (b) a feed-forward network which modifies each individual token embedding. The two main operations are applied along with residual connections and layer normalization. There may additionally be a position encoding incorporated into the embedding during self-attention stage as well.

To use Transformers as decoder-only language models, a popular paradigm is that of causal language modeling. Given a train dataset of examples each of which is a sequence of tokens of length $N$, causal language modeling simultaneously trains to minimize $N$ loss terms on each sequence. These loss terms minimize the cross-entropy loss of the model's prediction for token $\mathbf{t}^i$ using the prefix $\mathbf{t}^{1,\dots,i-1}$. During training, all $N$ of these loss terms can be evaluated in parallel with the use of causal masking. During decoding, the model iteratively generates one new token at a time by passing token $\mathbf{t}_i$ through the $\ell$ layers sequentially to obtain $\mathbf{t}_{i+1}$. This means that growing the network depth incurs a linear cost on the time taken to decode the next token during inference. However, there is ample evidence that depth is crucial for good quality models (Devlin et al., 2019; Raffel et al., 2023). There is fundamentally no way to avoid this cost in a Transformer, since every token relies on the completed predictions of every other prior token.

**StagFormer**   StagFormer introduces a way to break the sequential dependency of layers within a Transformer network and still be able to perform efficient and performant causal language modeling. We first partition our $\ell$ layers into $p$ sub-networks we call *stacks*. For ease of exposition we will first focus on the simplest case $p = 2$. Let $h = \ell/2$. StagFormer allows for execution of the stacks of layers $1, \dots, h$ and $h + 1, \dots, \ell$ in parallel in a given time step $i$ by *staggering* the dependency between $\mathbf{t}_h^i$ and $\mathbf{t}_{h+1}^i$.

In particular, we compute $\mathbf{t}_{h+1}^i$ as a function of the original token sequence $\mathbf{t}_0^{1,\dots,i}$ and the $h^{th}$ layer representations taken until time step $i - 1$: $\mathbf{t}_h^{1,\dots,i-1}$. Crucially we exclude a dependency on $\mathbf{t}_h^i$. This allows the lower half of layers to begin computing the predictions for the next token in the sequence, $\mathbf{t}_h^{i+1}$, while the upper layers in the network are finishing computing the final prediction for position $i$, $\mathbf{t}_\ell^i$.

We realize this by passing the original token embedding, $\mathbf{t}_0^i$ as input to the second half of the layers, $L_{h+1}, \dots, L_\ell$, and by augmenting these layers with cross attention to the final activations of the first half of the network on the prior tokens, $\mathbf{t}_h^1, \dots, \mathbf{t}_h^{i-1}$, when computing the final predictions for the next token after position $i$. Thus $\mathbf{t}_{h+1}^i$ does not actually depend on the prior layers' representation of the token, $\mathbf{t}_h^i$, it is a function of the initial token embedding, $\mathbf{t}_0^i$, and cross-attends to the previous layers' representations of only past tokens, $\mathbf{t}_h^0, \dots, \mathbf{t}_h^{i-1}$.

Figure 1 shows a timing diagram of how decoding works in StagFormer. The parallel execution of the two stacks is shown more clearly in Figure 2. During training, to faithfully simulate StagFormer's decoding, we sequentially pass our token sequence over the two stacks of layers where we allow the second stack to cross-attend to the outputs of the first stack with masking such that at position $i$ we can only cross-attend to the first $i - 1$ outputs from the first stack. This completes a description of how we can train and decode using StagFormer. The full algorithm is given is Algorithm 1.

This idea can be generalized to $p$ partitions of the $\ell$ layers by having each new partition stagger an additional time-step. We call this technique *staggering* the Transformer network over $p$ stacks. A full description of this generalization is presented in Section 4.3.

**Quantifying the Latency Benefits of StagFormer.**   The main advantage of StagFormer is the potential to save latency during decoding by executing stacks in parallel. This can be realized efficiently on today's hardware accelerators such as TPUs and GPUs. Staggering the dependency on the processed representations of tokens until time step $i$ between the first and second stacks of StagFormer can, in principle, lead to a decrease in quality of the model. However, the additional cross-attention parameters in the second stack help ameliorate this decline.

How much compute savings does StagFormer afford us? We first perform a theoretical analysis to understand the optimal savings we can hope to achieve.

Assume our baseline transformer has depth $\ell$ which is evenly divisible by 2. If the number of FLOPs in the embedding/unembedding layers are $e$ each and the number of MLP and

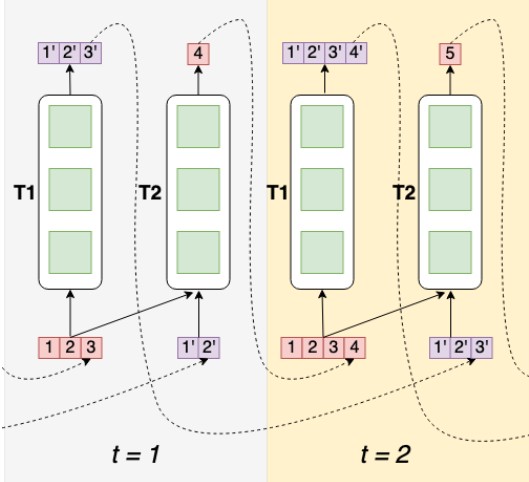

Figure 2: Depiction of the parallel execution of stacks T1 and T2 in a 2-stack StagFormer. In a given time step, both T1 and T2 can run in parallel: T1 producing the intermediate activation to be used in the next time step and T2 producing the output token for the next time step.

---

**Algorithm 1** StagFormer algorithm

---

**Input:** $\mathbf{t}_0^1, \ldots, \mathbf{t}_0^i \in \mathbb{R}^d$ : Token embeddings for positions $1, \ldots, i$ in the input sequence.

**Output:** $\mathbf{t}_\ell^i \in \mathbb{R}^d$ : The predicted token embedding for position $i + 1$ in the input sequence where $\ell$ is the total number of Transformer layers in the network.

1: **First pass** : for each layer $L_1, ..., L_h$ where $h \equiv \ell/2$ compute $\mathbf{t}_j^i = L_j \left( \mathbf{t}_{j-1}^{1,\ldots,i} \right)$.

Each application of $L_j$ using standard Transformer layer with self-attention and feed-forward layers.

2: **Second pass** : for each layer $L_{h+1}, \ldots, L_\ell$ compute $\mathbf{t}_j^i = L'_j \left( \mathbf{t}_u^{1,\ldots,i}, \mathbf{t}_h^{1,\ldots,i-1} \right)$.

Where $u = 0$ when $j = h + 1$ and $u = j$ otherwise.
Where $L'_j$ is a Transformer layer that has an additional cross-attention layer between the self-attention and feed-forward layers that uses $\mathbf{t}_h^{1,\ldots,i-1}$ for KV inputs.

3: **Return** $\mathbf{t}_\ell^i$

---

attention flops per transformer layer are $m, a$ respectively, then the total FLOPs of a forward pass through the baseline are $2e + \ell(m + a)$.

Now consider a StagFormer of depth $\ell$ with 2 stacks. The total number of FLOPs are $2e + 3\ell(m + a)/2$. However, due to the staggering we can execute the bottom stack and top stack in parallel. In modern accelerator hardware, when two computation graphs can be executed in parallel, the amount of time it takes to complete both of them depends on how busy we can keep the hardware. That is, how close to the **arithmetic intensity** of the hardware (which roughly measures how peak FLOPs/sec the hardware supports) the computation graph executes. This requires minimizing the communication time across chips and memory read/write time as well. In the ideal scenario when we are able to double the number of chips we have, we can achieve a latency equivalent to that of a model executing $2e + \ell(m + a)/2$ FLOPs.

In Section 3, we train and evaluate StagFormer for language modeling and observe that a depth $\ell$ StagFormer with 2 stacks outperforms a depth $\ell$ regular Transformer (Table 1). At the same time, we measure the potential latency speedup using a setup that simulates Stagformer with 2 stacks on twice the number of chips used by a baseline model. While we faithfully account for every segment of the StagFormer model, we ignore the inter-chip communication cost between the first and second stacks of the Stagformer. This communication cost is expected to be minimal in practice under an optimized hardware setup. We compare the time it takes StagFormer to generate text compared to a baseline Transformer model in Figure 3. Overall, we see strong performance gains on tasks such as SQuADv2, Lambada and HellaSwag while being neutral with the baseline on some others such as SuperGLUE.

Figure 3: Simulated Latency Benchmarking for a baseline Transformer (dotted) vs a comparable quality StagFormer model (solid). We plot the total time it takes to decode up to 2,048 tokens after a 8,192 token prefill. We observe that StagFormer is able to decode 2,048 tokens than baseline even with double the effective batch size.

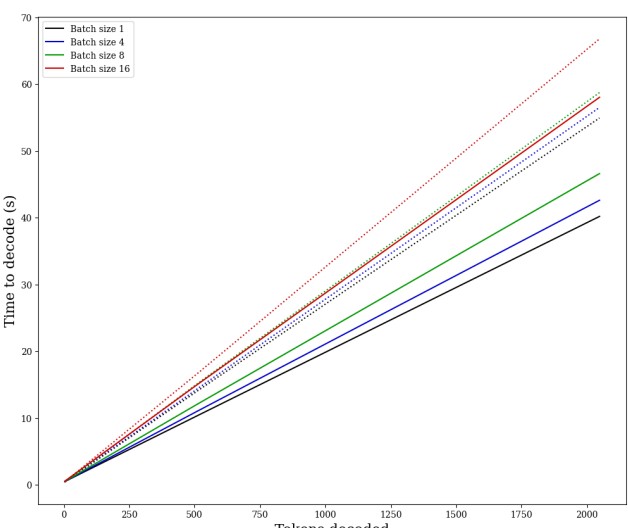

**Impact of StagFormer on Memory Usage** One limitation of StagFormer is that it increases the memory used by a Transformer's KV caches during decoding. This is due to the additional cross-attention layers that we add to the latter stack of the model which introduce a cross-attention KV cache in addition to the self-attention KV cache for the latter stack. This increases the total KV cache memory usage by precisely 50%. In Section 4.2 we discuss a variant of StagFormer where we use local attention in order to reduce the increase in KV cache size.

In the next section, we describe some extensions of the StagFormer idea which might be more applicable for specific settings.

## 3   Experiments

In this section, we describe our pre-training downstream evaluation setup we used for the different variants of the StagFormer via causal language modeling on the Pile dataset (Gao et al., 2020). We begin by outlining our experiment setting.

### 3.1   Experimental Setting

We performed our experiments using a standard Transformer architecture. The model uses a vocabulary size of 256,000. The model adds global positional embeddings to initial token embeddings and applies Rotary Positional Embeddings (RoPE) in the attention layers (Su et al., 2023). We compare StagFormer to an 18 layer baseline model with 1.6 billion parameters as well as a baseline where we double the number of layers, resulting in a 2.8 billion parameter model. We pretrained our model on The Pile dataset with a global batch size of 1024 and a max sequence length of 1280  (Gao et al., 2020). We trained the model for 250, 000 steps or  327 billion tokens which  Gu & Dao (2024) demonstrated should be enough tokens for the model to develop few-shot learning capabilities.

We evaluate the model's performance on several few-shot learning tasks  (Brown et al., 2020). The evaluation benchmarks include HellaSwag, ARC-E/C, WinoGrande, SuperGLUE, SQuADv2, and GEM-XSum  (Zellers et al., 2019; Ma et al., 2023; Sakaguchi et al., 2019; Wang et al., 2020; Rajpurkar et al., 2018).

### 3.2   Results

We first present latency benchmarking results on accelerator hardware which demonstrate the gains we are able to see during decoding with StagFormer compared to a quality matched standard Transformers. The analysis is presented in Table 3.

We compare a model with double the number of Transformer layers with a separate-weights StagFormer model which uses the same number of layers as the original baseline model in each pass. We chose to compare StagFormer to a Transformer with double the number of layers to compare the benefits of using staggered passes with adding more layers to the model. See Table 1 in the Introduction for results.

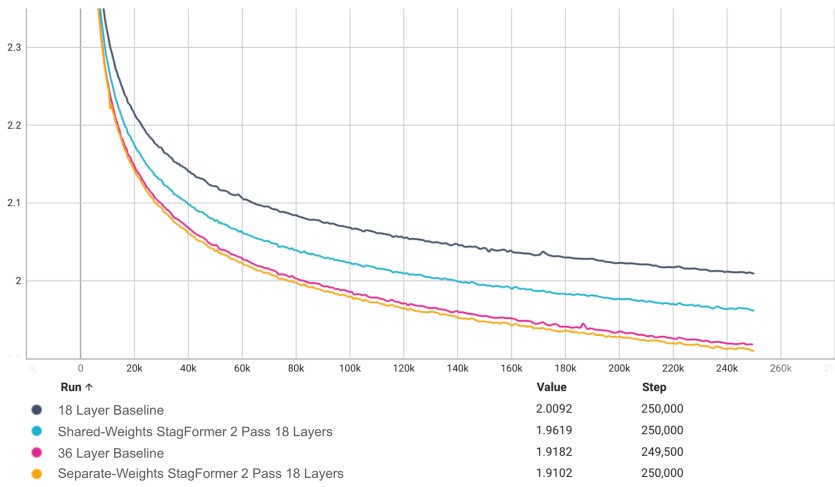

Figure 4: Plot of the training perplexity loss for the 18 layer baseline (black), 18 layer shared-weights StagFormer (blue), the 36 layer baseline (red), and separate-weights StagFormer with 2 stacks of 18 layers (yellow). We see that shared-weights StagFormer closes much of the gap between the 18 and 36 layer baselines. We also see that separate-weights StagFormer is matches and slightly surpasses the 36 layer baseline.

# 4 Extensions of the StagFormer

In this section, we describe certain natural extensions and variants of the StagFormer architecture.

## 4.1 Shared-Weights StagFormer

A simple variant of the StagFormer architecture is one where the two stacks share all weight parameters (except the cross-attention parameters). This can be useful in scenarios where we are bound tightly on memory requirements. Moreover, it allows us to execute both stacks on the same core using batching of the input data.

One limitation of shared-weights StagFormer is that it increases the KV cache size. A 2 stack shared-weights StagFormer will have triple the KV cache size of a Transformer model: StagFormer will need one KV cache for self-attention for each stack and one additional KV cache for the cross attention. In Section 4.2 we discuss a variant of StagFormer which uses local cross-attention to reduce the size of the KV cache. In Section A we discuss a variant of shared-weights StagFormer which reduces the KV cache size by simulating a recurrent neural network (RNN).

**Experimental Results**    At the 1-3 billion parameter scale, we compare shared-weights StagFormer to a baseline model with the same number of layers. The results with Shared-Weights StagFormer are presented in Table 2. We remark that a 2 stack shared-weights StagFormer with each stack having 18 layers performs significantly better than a 18 layer baseline model which has a similar number of parameters. Therefore, StagFormer is an effective way of boosting the performance given a fixed parameter budget (similar to Dehghani et al. (2018); Lan et al. (2019); Saunshi et al. (2025) and other related works).

Table 2: Performance of shared-weights StagFormer pretraining and inference. We see that shared-weights StagFormer is able to close much of the gap between the 18 and 36 layer baselines and is more parameter efficient than the 18 layer baseline model.

| Model | Pile Pplx. | HellaSwag | ARC-E | ARC-C | WinoGrande | SuperGLUE | SQuADv2 | GEM-XSum rouge2 | Avg. |
|---|---|---|---|---|---|---|---|---|---|
| Baseline (18L) 1.6B params | 4.026 | 49.8 | 60.1 | 31.8 | 53.4 | 59.3 | 31.8 | 0.9 | 41.0 |
| Baseline (36L) 2.8B params | 3.780 | 53.3 | 66.7 | 34.6 | 60.4 | 62.1 | 36.3 | 1.6 | 45.0 |
| StagFormer $p = 2$ Shared-Weights 18L 1.8B params | 3.896 | **54.3** | 61.7 | 31.7 | 57.7 | 59.5 | **46.9** | **2.1** | **44.8** |

## 4.2 StagFormer with Local Cross-Attention

If we want stronger latency savings, a further optimization for StagFormer that is simple to implement is to use local attention for the cross-attention between passes (Beltagy et al., 2020). Using local attention also reduces the additional memory we need to store in the KV cache. We observe that StagFormer still performs well even when using local cross-attention with relatively small attention window sizes. StagFormer is also capable of giving non-trivial quality when using an attention window size of 1, which converts the application of the cross-attention in layer $L_j$ on token $\mathbf{t}_{j-1}^i$ to a linear function of $\mathbf{t}_h^{i-1}$ (recall $h \equiv \ell/2$).

**Experimental Results**    We run experiments using StagFormer with local cross-attention with both the separate- and shared-weights variants. We present results for experiments with local attention using window sizes 512, 128, and 1. We remark that local cross-attention can be used successfully with both the separate-weights and shared-weights variants. See Table 4 and 5 in the Appendix for the results of our experiments using local attention with separate-weights and shared-weights respectively.

### 4.3 StagFormer with More Than Two Stacks

A natural extension of StagFormer idea we had touched upon earlier is to have $h$ be less than $\ell/2$ and to stagger over more than 2 stacks through the network. For instance, we could have $h \equiv \ell/3$ and stagger the network over 3 stacks. Let $p$ be the number of stacks we stagger the network over, then $h \equiv \ell/p$. Intuitively, as we increase the number of stacks $p$, due to progressive staggering, at time step $i$ stack $s$ only gets to see tokens until time step $i - p + s$ but needs to produce activations which help predict token $i + 1$. Thus the job becomes more difficult to learn as $p$ increases, and the depth of each stack reduces which contributes to eventual degradation in quality.

Our early experiments found that model quality suffers when $p > 2$. However, we find that we can recover significantly by imploring a simple change for StagFormer: rather than using just the final stack's output for computing the final logits, we use a linear combination of each stack's output with learnable coefficients, $\alpha_1, \ldots, \alpha_p$. Algorithm 4 defines separate-weights StagFormer for when $p > 2$ in the Appendix.

**Experimental Results**   We explored the settings of $p = 3, 4$ in this work. We find that as we increase $p$ model quality suffers, but there might be ways to extend the approach effectively to even larger values of $p$ which we leave for future work. Our experimental results for this variant are shown on Table 6 in the Appendix.

We also explore another variant of the shared-weights StagFormer - one which allows for a recurrent inference pathway in Section A in the Appendix. This variant has a very efficient decode step, however the forward pass between training and inference differs leading to slight regression on long generation tasks.

## 5   Conclusion

We present the StagFormer architecture as a way to increase the capacity of Transformer models by allowing lower-level layers to attend to the final activations produced by the same or different networks. With separate-weights StagFormer, we demonstrate that we can use higher level representations of prior tokens to run data-independent Transformer layers in parallel to process the current token without sacrificing quality.

**Future work and limitations**   We believe the StagFormer idea offers many interesting avenues for future research. For example, training shared-weights StagFormer only approximates recurrent inference, since training requires a discrete number of passes. Future work could explore how to extend the StagFormer algorithm that either better approximates or fully realizes recurrent decoding with better quality.

Another limitation is that cross-attention incurs additional computational cost in both time and space with respect to the input length. In the separate-weights StagFormer variant, cross attention increases the KV cache size by 50%. Instead of passing the output from the first stack as an extra input to attend to, if we could somehow merge it with the token embeddings then we would remove the additional cross-attention cost and KV cache. Such a design would greatly increase efficiency.

The shared-weights variant of StagFormer also increases the KV cache size. Because of the additional pass with self- and cross-attention in the second pass, shared-weights Stagformer triples the KV cache size relative to a traditional Transformer with the same number of parameters.

Future work can explore alternate ways to reuse information-rich higher level activations in lower-level layers to allow parallel execution of layers in a way that incurs less computational cost than attention and matches a deeper model's quality.

One material challenge of StagFormer's parallel execution of layers is that it would require a communication cost to copy the result from one network over to the other. This prevents one from realizing the full theoretical latency benefit of running the StagFormer towers in parallel. Furthermore, since most models rely on the single program, multiple data (SPMD)

paradigm (Xu et al., 2021), parallel execution of StagFormer stacks would require storing a copy of the token embeddings and final softmax tables in both cores when executing StagFormer stacks. Further work could explore how to extend this algorithm or the SPMD paradigm to help realize greater latency benefits when executing Transformer networks in parallel.

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

# A  A Recurrent Approximation to the Shared-Weights StagFormer

Consider the shared-weights StagFormer with $p = 2$. It suggests the following variant which we will see can have many benefits. We perform the first pass prefill on the input tokens as before storing the final activations. During decoding, rather than having the network process each token twice, we only do a single pass. When doing so, the network cross-attends to the final activations of all prior tokens including the prefill tokens and generated tokens so far.

This method of decoding resembles a recurrent neural network (RNN) where the final activations of prior tokens are the RNN's hidden state and cross-attention serves as a gating mechanism while processing the current token. Though one key difference is that a traditional RNN's hidden state remains a fixed size, but shared-weights StagFormer grows the size of its "hidden state" of upper layer activations as the sequence length grows.

This method reduces the size of the additional KV cache needed for StagFormer. Recall that a 2 pass shared-weights StagFormer increases the KV cache size because it needs additional caching for both the self- and cross-attention layers. The recurrent StagFormer variant, on the other hand, only needs to add additional KV cache for cross-attention and does not require any additional caching for the self-attention layers.

We show that it is possible to use shared-weights StagFormer for recurrent decoding using this scheme, even when the model is trained using two separate passes. However, we find that the generated text's quality is not as good as when we process decode new tokens the original way, with two networks running in parallel.

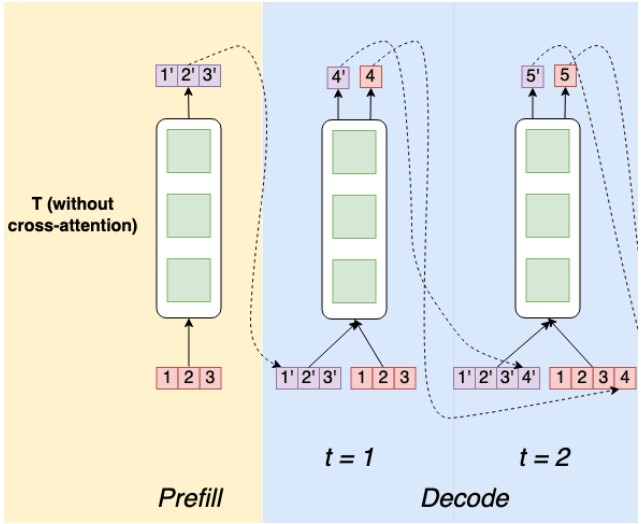

Figure 5: Timing Diagram of Prefill vs Decode steps for Recurrent Inference with Shared-Weights StagFormer. During prefill, the Transformer T is run without cross-attention and during decode it is run with cross-attention.

Table 3: Performance of Shared-Weight StagFormer pretraining and recurrent inference using Shared-Weight StagFormer. We see that the recurrent variant regresses in sampling tasks like SQuADv2 and GEM-XSum.

| Model | Pile Pplx. | HellaSwag | ARC-E | ARC-C | WinoGrande | SuperGLUE | SQuADv2 | GEM-XSum rouge2 | Avg. |
|---|---|---|---|---|---|---|---|---|---|
| Baseline (18L) 1.6B params | 4.026 | 49.8 | 60.1 | 31.8 | 53.4 | 59.3 | 31.8 | 0.9 | 41.0 |
| Baseline (36L) 2.8B params | 3.780 | 53.3 | 66.7 | 34.6 | 60.4 | 62.1 | 36.3 | 1.6 | 45.0 |
| StagFormer $p = 2$ Shared-Weights 18L Two-Networks 1.8B params | 3.896 | 54.3 | 61.7 | 31.7 | 57.7 | 59.5 | 46.9 | 2.1 | 44.8 |
| StagFormer $p = 2$ Shared-Weights 18L Recurrent 1.8B params | 3.896 | 54.3 | 61.7 | 31.7 | 57.7 | 59.5 | **42** | **0.4** | 43.9 |

**Recurrent StagFormer with More Than Two Passes**  One can also increase the number of staggered passes with shared-weights StagFormer when training to better approximate recurrent decoding. Since the Transformer layer weights are shared between passes, shared-weights StagFormer would process the same input multiple times, cross-attending to prior tokens' final activations from prior passes.

For shared-weights StagFormer, we match training during prefill and run all $p$ stacks, and then switch to recurrent inference for decoding. Note that for $p = 4$, some evaluation tasks failed due to memory constraints. We find that increasing $p$ surprisingly has a negative

impact on model quality. See Table 7 in the Appendix for results using $p = 2, 3, 4$ for the recurrent variant of shared-weights StagFormer.

## B  Additional Details on StagFormer Extensions and Experiments

We begin by presenting the full algorithm for the shared-weights variant of the StagFormer in Algorithm 2.

---

**Algorithm 2** Shared-weights StagFormer algorithm

---

    **Input:** $\mathbf{t}_0^1, \ldots, \mathbf{t}_0^i \in \mathbb{R}^d$ : Token embeddings for positions $1, \ldots, i$ in the input sequence.

    **Output:** $\mathbf{t}_l^i \in \mathbb{R}^d$ : The predicted token embedding for position $i + 1$ in the input sequence where $l$ is the total number of Transformer layers in the network.

1: **First pass** : for each layer $L_1, ..., L_l$ compute $\mathbf{t}_j^i = L_j\left(\mathbf{t}_{j-1}^{1,\ldots,i}\right)$.

    Each application of $L_j$ using standard Transformer layer with self-attention and feed-forward layers.

2: **Second pass** : for each layer $L_1, \ldots, L_l$ compute $\mathbf{t}_j^i = L_j'\left(\mathbf{t}_{j-1}^{1,\ldots,i}, \mathbf{t}_L^{1,\ldots,i-1}\right)$.

    Where $L_j'$ has an additional cross-attention layer between the self-attention and feed-forward layers to the Transformer layers in the first pass that uses $\mathbf{t}_l^{1,\ldots,i-1}$ for KV inputs.

3: **Return** $\mathbf{t}_l^i$.

---

Next, we present how with a small modification to Algorithm 2 we can approximate a recurrent model in Algorithm 3.

---

**Algorithm 3** Recurrent Decoding using Shared-Weights StagFormer

---

    **Input:** $\mathbf{t}_0^1, \ldots, \mathbf{t}_0^i \in \mathbb{R}^d$ : Token embeddings for positions $1, \ldots, i$ in the input sequence.

    **Output:** $\mathbf{t}_\ell^i \in \mathbb{R}^d$ : The predicted token embedding for position $i + 1$ in the input sequence where $L$ is the total number of Transformer layers in the network.

1: **Prefill** : Use the shared-weights StagFormer algorithm to process the prefix (Algorithm 2).

2: **Decoding** : for each layer $L_1, \ldots, L_\ell$ compute $\mathbf{t}_j^i = L_j'\left(\mathbf{t}_{j-1}^{1,\ldots,i}, \mathbf{t}_l^{1,\ldots,i-1}\right)$.

    Where $L_j'$ has an additional cross-attention layer between the self-attention and feed-forward layers to the Transformer layers in the first pass that uses $\mathbf{t}_l^{1,\ldots,i-1}$ for KV inputs. The rest of the parameters in $L_j'$ are the same as those in $L_j$ used for the prefill.

3: **Return** $\mathbf{t}_\ell^i$

---

The last algorithm we present is the one for extending the StagFormer idea to more than 2 stacks (Algorithm 4).

We also present experimental results for both the separate- and shared-weights variants StagFormer with local cross-attention.

Finally, we present experimental results for using StagFormer with more than 2 passes, or in other words, when $p > 2$.

---

**Algorithm 4** Separate-weights StagFormer $p > 2$ algorithm

---

**Input:** $\mathbf{t}_0^1, \ldots, \mathbf{t}_0^i \in \mathbb{R}^d$ : Token embeddings for positions $1, \ldots, i$ in the input sequence.

**Output:** $\mathbf{t}_\ell^i \in \mathbb{R}^d$ : The predicted token embedding for position $i + 1$ in the input sequence where $\ell$ is the total number of Transformer layers in the network.

1: **First pass** : for each layer $L_1, \ldots, L_h$ where $h \equiv \ell / p$ compute $\mathbf{t}_j^i = L_j \left( \mathbf{t}_{j-1}^{1,\ldots,i} \right)$.

Each application of $L_j$ using standard Transformer layer with self-attention and feed-forward layers.

2: **Subsequent passes** : for each $k \in \{2, \ldots, p\}$ do:

for each layer in $L_{h \cdot (k-1)+1}, \ldots, L_{h \cdot k}$ compute $\mathbf{t}_j^i = L_j' \left( \mathbf{t}_u^{1,\ldots,i}, \mathbf{t}_{h \cdot (k-1)}^{1,\ldots,i-1} \right)$.

Where $u = 0$ when $j = h \cdot (k-1) + 1$ and $u = j$ otherwise.

Where $L_j'$ is a Transformer layer that has an additional cross-attention layer between the self-attention and feed-forward layers that uses $\mathbf{t}_{h \cdot (k-1)}^{1,\ldots,i-1}$ for KV inputs.

3: **Return** $\sum\limits_{k}^{p} \alpha_k \cdot \mathbf{t}_{h \cdot k}^i$ .

Where each $\alpha_k$ is a learnable scalar.

---

Table 4: Performance of separate-weights StagFormer on pretraining and eval tasks with local cross-attention. We see that StagFormer is able to maintain its quality when the window size is 512. However, we see when we reduce the cross-attention window size further that quality begins to degrade.

| Model | Pile Pplx. | HellaSwag | ARC-E | ARC-C | WinoGrande | SuperGLUE | SQuADv2 | GEM-XSum rouge2 | Avg. |
|---|---|---|---|---|---|---|---|---|---|
| Baseline 2x Layers (36L) 2.8B params | 3.780 | 53.3 | 66.7 | 34.6 | 60.4 | 62.1 | 36.3 | 1.6 | 45.0 |
| StagFormer Separate-Weights Window 512 2.9B params | 3.767 | **58.6** | **68.2** | **36.9** | **61.8** | **63.3** | **41.5** | **1.9** | **47.5** |
| StagFormer Separate-Weights Window 128 2.9B params | 3.797 | 51.3 | 55.6 | 32.8 | 59.6 | 59.1 | 21.5 | 1.1 | 40.1 |
| StagFormer Separate-Weights Window 1 2.9B params | 3.818 | 33.3 | 30.9 | 25.3 | 51.2 | 45.6 | 0 | 0 | 26.6 |

Table 5: Performance of shared-weights StagFormer on pretraining and eval tasks with local cross-attention. We see that shared-weights StagFormer is able to maintain their quality with cross-attention window sizes of 512 and 128. However, we see that when the window size is 1 that quality begins to suffer.

| Model | Pile Pplx. | HellaSwag | ARC-E | ARC-C | WinoGrande | SuperGLUE | SQuADv2 | GEM-XSum rouge2 | Avg. |
|---|---|---|---|---|---|---|---|---|---|
| Baseline 18L 1.6B params | 4.026 | 49.8 | 60.1 | 31.8 | 53.4 | 59.3 | 31.8 | 0.9 | 41.0 |
| StagFormer Shared-Weights Window 512 1.8B params | 3.908 | **55.7** | 64.9 | 33.9 | 59.4 | 60.1 | **39.4** | 1.6 | **45.0** |
| StagFormer Shared-Weights Window 128 1.8B params | 3.929 | **56.4** | 64.9 | 34 | 59.4 | 59.8 | **40.3** | **1.8** | **45.2** |
| StagFormer Shared-Weights Window 1 1.8B params | 3.951 | 46.8 | 56.5 | 29.4 | 58.5 | 58 | 34.8 | 0.6 | 40.7 |

Table 6: Performance of StagFormer on pretraining and a subset of evaluation tasks when $p > 2$. In this case, we need the output of StagFormer to be a linear combination of each stack's output in order to not degrade quality. For $p = 3$ we see the model quality does not degrade much except for a significant regression in SQuADv2.

| Model | Train Pplx. | HellaSwag | ARC-E | ARC-C | WinoGrande | SuperGLUE | SQuADv2 | GEM-XSum rouge2 | Avg. |
|---|---|---|---|---|---|---|---|---|---|
| Baseline 18L 1.6B params | 4.026 | 49.8 | 60.1 | 31.8 | 53.4 | 59.3 | 31.8 | 0.9 | 41.0 |
| Baseline 2x Layers (36L) 2.8B params | 3.780 | 53.3 | 66.7 | 34.6 | 60.4 | 62.1 | 36.3 | 1.6 | 45.0 |
| StagFormer $p = 3$ Separate-Weights (3 x 12L) 3.0B params | 3.766 | 52.9 | 52.7 | 29.1 | 55.2 | 60 | 13.7 | 1 | 37.8 |
| StagFormer $p = 4$ Separate-Weights (4 x 9L) 3.0B params | 3.797 | 51.3 | 58 | 30.5 | 55 | 59.3 | 33.1 | 1.2 | 41.2 |

Table 7: Performance of Shared-Weights on pretraining and recurrent inference StagFormer on a subset of evaluation tasks when $p \geq 2$. For $p = 4$ some evaluation tasks crashed due to memory constraints. We find that as we increase $p$, the model does not better simulate an RNN and quality degrades.

| Model | Train Pplx. | HellaSwag | ARC-E | ARC-C | WinoGrande | SuperGLUE | SQuADv2 | GEM-XSum rouge2 | Avg. |
|---|---|---|---|---|---|---|---|---|---|
| Baseline 18L 1.6B params | 4.026 | 49.8 | 60.1 | 31.8 | 53.4 | 59.3 | 31.8 | 0.9 | 41.0 |
| StagFormer $p = 2$ Shared-Weights 18L Recurrent 1.8B params | 3.896 | 54.3 | 61.7 | 31.7 | 57.7 | 59.5 | 42 | 0.4 | 43.9 |
| StagFormer $p = 3$ Shared-Weights 18L Recurrent 1.8B params | 3.858 | 51.3 | 55.6 | 31.8 | 59.6 | 59.1 | 21.5 | 1.1 | 40.0 |
| StagFormer $p = 4$ Shared-Weights 18L Recurrent 1.8B params | 3.870 | 46.6 | – | – | 51.9 | – | 5 | 0.6 | 26.0 |

