# OpenReview forum: "StagFormer: Time Staggering Decoder only Transformers"
_colmweb.org/COLM/2025/Conference — COLM 2025_

### Official Review · Reviewer_TYAW · 2025-04-27

**Rating:** 7
**Confidence:** 3
**Ethics Flag:** 1

**Summary:**

This paper proposes StagFormer, a method to improve the decoding efficiency through introducing some degrees of parallelization. In StagFormer, transformers can be divided into 2 stacks with different layers (lower half and upper half, though weights can be shared). The representations from the lower half can be passed into the upper half by skipping time steps, therefore achieving multi-step decoding. Experiments show that StagFormer can improve efficiency while maintaining the same performance as vanilla transformers. The authors also discuss many different implementations of StagFormer.

**Questions To Authors:**

1. The authors only considers l/2 case where the upper half and the lower half are equal in weights. I know this facilitates weight sharing. If the two halves are unequal (like a heavy lower half and a lightweight upper half), how will the performance change? Medusa only uses very lightweight decoding heads at the last layers. It seems that equal weights are not always necessary.

**Reasons To Accept:**

1. The proposed StagFormer can be considered novel, building on existing works. The recurrent formulation is particularly interesting and can inspire future works.
2. The improvement over vanilla transformers seems pretty well established, given the experiments.

**Reasons To Reject:**

1. Lack of comparison. The authors only compare the proposed method with vanilla transformers. Yet there are already many improvements over the vanilla transformers. It could help the readers better contextualize the results with some well-established baselines, such as Medusa and speculative decoding.
2. Result presentations. There are some issues with the result presentations.
    - The main contribution of this paper is an efficient decoding method. Yet, the efficiency performance is only presented in Table 2 for 1024 tokens for the separate-weight variant. It will be more helpful if the efficiency can be presented across sequence length?
    - The authors mention that StagFormer without local attention can increase KV cache. How much more compared to a vanilla performance? Quantifying this trade-off can help better illustrate the performance of StagFormer.
    - The labels for StagFormer variants seem a bit inconsistent.  In the Appendix, variants are labeled either as shared-weights or separate-weights. In Table 1, I am assuming that the StagFormer is a separate-weight variant, though it is not labeled explicitly.

---

> ### Author Response · Authors · 2025-06-03
> **Response to Reviewer TYAW**
>
> We thank you for your careful consideration and review of our work. Below let us address each of the points you brought up:
>
> > The authors only compare the proposed method with vanilla transformers. Yet there are already many improvements over the vanilla transformers. It could help the readers better contextualize the results with some well-established baselines, such as Medusa and speculative decoding.
>
> While we agree it is natural to compare StagFormer to other Transformer efficiency techniques such as Medusa or speculative decoding, we did not think this comparison was in scope for this paper. Speculative decoding is a technique which can be employed with or without StagFormer. Likewise, Medusa could be employed using a StagFormer network as the Original Model component of the Medusa setup. These techniques are improving inference latency by optimizing different aspects of the network. For these reasons, we consider these techniques to be orthogonal to StagFormer and did not include comparisons in this introductory work.
>
> > The main contribution of this paper is an efficient decoding method. Yet, the efficiency performance is only presented in Table 2 for 1024 tokens for the separate-weight variant. It will be more helpful if the efficiency can be presented across sequence length?
>
> The latency figure in Table 2 displays latency figures for a 1,024 token prefill and then an average latency per token after decoding 1,024 tokens autoregressively. We can add an additional study of latency as sequence length varies in our final manuscript.
>
> > The authors mention that StagFormer without local attention can increase KV cache. How much more compared to a vanilla performance? Quantifying this trade-off can help better illustrate the performance of StagFormer.
>
> This is an excellent point. We can update  the "Future work and limitations" section with a quantification of the impact StagFormer has on the KV cache size in the final revision.
>
> > The labels for StagFormer variants seem a bit inconsistent. In the Appendix, variants are labeled either as shared-weights or separate-weights. In Table 1, I am assuming that the StagFormer is a separate-weight variant, though it is not labeled explicitly.
>
> Thank you for the feedback, and I see how this has caused some confusion. We will update the labeling in Table 1 in the final revision to reflect that the results displayed there are for the separate-weights variant of Stagformer.
>
> > The authors only consider the l/2 case where the upper half and the lower half are equal in weights. I know this facilitates weight sharing. If the two halves are unequal (like a heavy lower half and a lightweight upper half), how will the performance change? Medusa only uses very lightweight decoding heads at the last layers. It seems that equal weights are not always necessary.
>
> This is an interesting point. A large focus of our work on StagFormer is on running the two halves of the network in parallel, and, for practical purposes, it is easier to run halves that are the same size across multiple devices. Additionally if we have imbalanced stacks running in parallel, then the latency will be bottlenecked by the largest of the stacks. Therefore having balanced stacks gives the optimal latency tradeoff. For this reason, we did not choose to focus on experimenting with imbalanced passes.
>
> Thank you again for all of your thoughtful feedback, and we look forward to any future discussions with you about our work.

---

> > ### Comment · Reviewer_TYAW · 2025-06-09
> >
> > Hi authors, thank you for your responses! I think your comments have addressed many of my concerns. I think the general idea of the paper is great but the presentation can still benefit from a bit more revision. If the authors agree to make changes as stated, I don't mind inceasing the score.

---

> > > ### Author Response · Authors · 2025-06-10
> > > **Response to Reviewer TYAW**
> > >
> > > Thank you for your response. We will make all the discussed changes and improve the presentation of the paper.

---

### Official Review · Reviewer_n1xq · 2025-05-13

**Rating:** 6
**Confidence:** 5
**Ethics Flag:** 1

**Summary:**

This paper introduces StagFormer, a Transformer variant with a direct and simple yet original idea for reducing decoding latency by staggering execution across model depth. This ambitious approach shows promising experimental results in latency reduction, and its "Recurrent Approximation" variant offers insightful avenues. However, the significance of these findings is somewhat tempered by relatively thin experimental validation across broader conditions. Additionally, the clarity of its core mechanism is impacted by a confusing Figure 1. While the concept is of good quality and addresses an important problem, the work would benefit from more robust empirical backing and clearer visual explanations to fully realize its potential impact.

**Reasons To Accept:**

mentioned in Summary

**Reasons To Reject:**

1. The figure 1 is confusing: My understanding is that at t=2, the prediction of the fifth token still depends on the output of the fourth token at t=1. If so, there is no latency improvement. figure 2 is much more clear, I think these two figures are duplicate.
2. The experiments are too thin: I suggest adding some comparisons with speculative sampling.

---

> ### Author Response · Authors · 2025-06-03
> **Response to Reviewer n1xq**
>
> We thank the reviewer for the positive feedback about our technique. Let us address each of your points of feedback.
>
>
> >  The figure 1 is confusing: My understanding is that at t=2, the prediction of the fifth token still depends on the output of the fourth token at t=1.
>
>
> Apologies for any confusion. Like a standard Transformer network, StagFormer only decodes one timestep at a time. Our diagram was to show the two stacks of Transformer layers, T1 and T2, can be executed in parallel in Stagformer (Fig 1 right) since they have no data dependency on each other at a particular timestep. This is in contrast with a standard Transformer network (Fig 1 left), which must execute all of its layers serially for each token.
>
>
> > [F]igure 2 is much more clear, I think these two figures are duplicate.
>
>
> In Figure 2, T1 and T2 are running in parallel during decoding to generate 1 token at a time. We understand if this is confusing since in Figure 1 we had both T1 and T2 aligned vertically when running in parallel for the same time step. While the figures are similar, the purpose of Figure 1 is to contrast the timing diagram of StagFormer with a traditional Transformer network, whereas Figure 2 is to show the timing diagram of StagFormer during generative inference.
>
>
> > The experiments are too thin: I suggest adding some comparisons with speculative sampling.
>
>
> Thank you for your feedback. While it seems natural to compare StagFormer with other transformer efficiency techniques like speculative decoding, we think this type of comparison is not in scope for this paper. Speculative decoding can be employed with or without the StagFormer architecture and vice versa. For this reason, we view StagFormer to be orthogonal to techniques like Medusa or speculative decoding.
>
>
> Thank you again for your review, and we look forward to participating in future discussion with you.

---

### Official Review · Reviewer_UGau · 2025-05-14

**Rating:** 7
**Confidence:** 3
**Ethics Flag:** 1

**Summary:**

This paper proposes a staggered layer formulation for a transformer, specially applicable to causal autoregressive decoders,  that allows to break the causal dependency between chunks of the model, allowing their parallel execution in a decoding phase while introducing little to no overhead in training and prefill phase. Experimental results show competitive performance in quality and better generation speed.

**Questions To Authors:**

* The Stagformer models seem to have a 0.1B params extra for each "P-1". That is, 2.9B for p=2, 3.0B for p=3. Did you try adding a few extra params to the baseline (ex. +1 transformer block) to compensate for this?

* The benefit of stagformer seems to come mostrly from parallelization of decoding. However, in many real world scenarios, inference can be batched to increase efficiency, and under that setting, the model becomes memory bound. It would be interesting to benchmark the model in such cases (batched inference) and measure the relative performance. There might still be benefits due to reused KV cache, but gains might diminish. Did you consider this?

* A clarification could be added in section 3.2 for the RNN parallelism: the key defining feature of an RNN state is its fixed (or, in some cases, bounded) size. A transformer is usually not referred to as an RNN because the state grows with the sequence. Could you please distinguish this case when drawing the analogy?

**Reasons To Accept:**

* Novel transformer variant with enhanced decoding parallelization at no quality cost and performance improvements in inference.
* Theoretically sound motivation with good results in practice
* Sufficent experimental results to back claims and sufficiently detailed description to allow independent reproduction.

**Reasons To Reject:**

* Some portions of the manuscript seem incomplete (ex. "TODO" annotations)
* Lack of considerations for real world deployment of the proposed decoding acceleration (batched inference vs single sequence)
* Explanation of theoretical inference speedup could be more detailed, without over-reliance on computational experiments to prove a speedup (on TPU, the operations are compiled from jax and it is unclear what computational graph (ex. loop unrolling, etc) is actually happening)

---

> ### Author Response · Authors · 2025-06-03
> **Response to Reviewer UGau**
>
> We thank the reviewer for their time and for their thoughtful comments. Let us address the points you brought up below:
>
> > Some portions of the manuscript seem incomplete (ex. "TODO" annotations)
>
> We made another pass over the paper but could not find any incomplete sentences or TODO annotations. We are happy to fix any such issues in the revision if the reviewer points out specific places where they occur.
>
> > Lack of considerations for real world deployment of the proposed decoding acceleration (batched inference vs single sequence)
>
> While not explicitly discussed in the paper, we posit that StagFormer offers decoding acceleration benefits in both the batched inference and single sequence cases. See response to the detailed question below.
>
> > Explanation of theoretical inference speedup could be more detailed, without over-reliance on computational experiments to prove a speedup (on TPU, the operations are compiled from jax and it is unclear what computational graph (ex. loop unrolling, etc) is actually happening)
>
> The shared-weights variant of StagFormer can realize its inference speedup by doubling the effective batch size of the input and processing each half of the new batch size on separate devices. We admit that we have omitted the cost of interdevice communication as a result of this sharding. For separate-weights StagFormer, we can achieve a similar setup but must broadcast the separate model weights to each device, which proved to be an implementation challenge for us. We will include a more detailed discussion on theoretical speedups in the revision.
>
> > The Stagformer models seem to have a 0.1B params extra for each "P-1". That is, 2.9B for p=2, 3.0B for p=3. Did you try adding a few extra params to the baseline (ex. +1 transformer block) to compensate for this?
>
> That is an excellent observation. We did not try to add additional parameters to a baseline model for this purpose. While it is possible that this difference in model size may have impacted the comparison of separate-weights StagFormer, we think the results with shared-weights StagFormer show that this technique provides benefit to performance beyond the additional few parameters we added.
>
> > The benefit of stagformer seems to come mostly from parallelization of decoding. However, in many real world scenarios, inference can be batched to increase efficiency, and under that setting, the model becomes memory bound. It would be interesting to benchmark the model in such cases (batched inference) and measure the relative performance. There might still be benefits due to reused KV cache, but gains might diminish. Did you consider this?
>
> Indeed, batching of inputs can make the model more compute bound and change the amount of inference speedup that Stagformer can offer. Another reviewer has asked us to compare latency between StagFormer and a standard Transformer when we vary sequence length, varying batch size may also provide useful insights as well. We will include a detailed comparison in the revision. That said, there are also many applications where batch size of one is unavoidable, in which case Stagformer should provide the largest speedups.
>
> > A clarification could be added in section 3.2 for the RNN parallelism: the key defining feature of an RNN state is its fixed (or, in some cases, bounded) size. A transformer is usually not referred to as an RNN because the state grows with the sequence. Could you please distinguish this case when drawing the analogy?
>
> Again, this is an excellent observation. We will update the text in the final revision to mention this difference between the recurrent StagFormer and more traditional RNNs.
>
> Thank you again for your review and for your insights into StagFormer, and we look forward to continuing our discussion with you in the future.

---

### Decision · Program_Chairs · 2025-07-08

**Decision:**

Accept

**Comment:**

StagFormer introduces an interesting staggered‐layer transformer variant that breaks causal dependencies between layer chunks to enable parallel decoding with minimal training or prefill overhead. The paper is technically sound and the proposed architecture is novel.

Reviewers UGau and TYAW both highlight its theoretically sound motivation and demonstrate strong empirical speedups and quality retention compared to vanilla transformers, with UGau praising its novel parallelization and TYAW confirming its well‐established inference improvements and clear recurrent formulation.

The paper also has several areas needing attention. UGau highlights the lack of clarity of writing, and the absence of real‐world deployment analysis—particularly under batched inference—and requests a more detailed theoretical account of the promised speedup and an equitable, parameter‐matched baseline comparison. TYAW recommends adding established decoding acceleration baselines like Medusa and speculative decoding, broadening latency studies across sequence lengths, explicitly labeling StagFormer variants, and quantifying the KV‐cache overhead. Addressing these points will greatly bolster the paper’s clarity and practical relevance.

Overall, I think it's a good contribution to the community of improving the efficiency of the current LLM paradigm.